# Benchmarking Object Detection Deep Learning Models in Embedded Devices

**DOI:** 10.3390/s22114205

**Published:** 2022-05-31

**Authors:** David Cantero, Iker Esnaola-Gonzalez, Jose Miguel-Alonso, Ekaitz Jauregi

**Affiliations:** 1TEKNIKER, Basque Research and Technology Alliance (BRTA), 20600 Eibar, Spain; iker.esnaola@basf.com; 2Department of Computer Architecture and Technology, University of the Basque Country UPV/EHU, 20018 San Sebastian, Spain; j.miguel@ehu.eus; 3Department of Languages and Information Systems, University of the Basque Country UPV/EHU, 20018 San Sebastian, Spain; ekaitz.jauregi@ehu.eus

**Keywords:** object detection, embedded devices, deep learning, benchmarking

## Abstract

Object detection is an essential capability for performing complex tasks in robotic applications. Today, deep learning (DL) approaches are the basis of state-of-the-art solutions in computer vision, where they provide very high accuracy albeit with high computational costs. Due to the physical limitations of robotic platforms, embedded devices are not as powerful as desktop computers, and adjustments have to be made to deep learning models before transferring them to robotic applications. This work benchmarks deep learning object detection models in embedded devices. Furthermore, some hardware selection guidelines are included, together with a description of the most relevant features of the two boards selected for this benchmark. Embedded electronic devices integrate a powerful AI co-processor to accelerate DL applications. To take advantage of these co-processors, models must be converted to a specific embedded runtime format. Five quantization levels applied to a collection of DL models are considered; two of them allow the execution of models in the embedded general-purpose CPU and are used as the baseline to assess the improvements obtained when running the same models with the three remaining quantization levels in the AI co-processors. The benchmark procedure is explained in detail, and a comprehensive analysis of the collected data is presented. Finally, the feasibility and challenges of the implementation of embedded object detection applications are discussed.

## 1. Introduction

Deep Learning (DL) is a sub-field of Machine Learning (ML) based on the computation of multi-layer Artificial Neural Networks (ANN), also known as Deep Neural Networks (DNN) in reference to the presence of multiple internal processing layers. One of the applications where DL is proving most successful is computer vision, where impressive levels of performance are being achieved. This work discusses object detection technology, which is defined as a computer vision technique that enumerates the objects presented in an image and classifies each of the detected objects, assigning a confidence or probability of existence while locating them and squaring their position in the image. In the traditional computer vision approach, object detection algorithms were based on handcrafted sets of features explicitly programmed by the authors. However, an object may present a diversity of morphological appearances and could be deformed, present a large variety of shapes and/or be immersed in scenes with very different illumination levels and backgrounds. Furthermore, objects may be partially occluded by other objects, making it almost impossible to extract robust features manually. DL, on the other hand, uses a huge amount of detection examples and trains a DNN to automatically infer the appropriate detection features. This strategy has proven to be highly successful.

Even if DL is a computationally intensive task, modern embedded hardware devices are powerful enough to execute some of the most successful models. In addition, hardware manufacturers have developed powerful AI (Artificial Intelligence) co-processors, specifically designed to execute DL models. These co-processors provide considerable computing power with high power efficiency. As a result, more and more AI-based applications are implemented in smart embedded devices [1]. Many techniques have been developed to improve the deployment of DL models on such devices, starting from simplified training processes using pre-trained networks and fine-tuning the parameters in a process called Transfer Learning [2], to many model simplifications and transformations, such as quantization, model pruning, etc., to squeeze the model onto embedded devices [3]. Note that even if the models are executed on the embedded devices, all the previous stages in the DL workflow cited above take place in powerful host computers, usually equipped with dedicated high performance graphics processing units (GPUs).

Embedded devices are of paramount importance to bring DL capabilities to robotic applications [4]. To name just a few examples, in [5] the authors present a system that can detect and track multiple objects from aerial images taken by a flying robot, while in [6] a 3D-printed robotic arm is brain-controlled via embedded DL from sEMG sensors. Real-time human detection is an important sub-field of computer vision, of interest in areas ranging from industrial environments to autonomous driving. For a review of this task using DL on embedded platforms, the reader is referred to [7].

The goal of this article is to provide a review of the major challenges in the development of embedded DL applications. The article is divided into two main parts. The first part presents a detailed analysis of the main elements to be taken into account in any DL embedded application: Section 2 explains the motivation for the use of embedded hardware and the most important features to be taken into account when selecting embedded devices. A description of the devices chosen for this work is also included. In Section 3, ML framework requirements are evaluated for both embedded hardware devices and host computers. The embedded hardware libraries are intended to provide a specific runtime environment for the execution of inference based on DL models in specialized hardware co-processors. ML host frameworks, on the other hand, are usually powerful software packages designed to support the whole DL application development workflow. Since the compatibility of both frameworks is mandatory, only a few options are feasible, so the selection is, as explained, quite straightforward. Section 4 describes some of the most successful and modern object detection models available and how they are handled by the selected ML framework.

The second part of the article carries out a benchmark of embedded hardware platforms based on the ML framework and previously identified models. Each model must be converted from its original format to an embedded-friendly format. Hardware co-processors support INT8 arithmetic operations, so model conversion also involves some kind of model quantization. Five quantization levels are considered for this work, as described in Section 5. After conversion, models are deployed in the embedded devices, and their inference performance is measured and tested. Section 6 describes the benchmark procedure and analyzes the obtained results. Finally, Section 7 states the conclusions of this work, and Section 8 enumerates some reflections about future lines of work.

## 2. AI at the Edge: Intelligent Embedded Systems

Edge computing is a distributed computing architecture where most data processing is executed by hardware devices close to the source of the data. As opposed to cloud computing, where large and powerful central facilities receive huge amounts of data from remotely connected sensors and compute complex and performance-demanding algorithms, edge computing brings the computation to devices with limited resources.

Related to cloud computing, the Internet of Things (IoT) paradigm, which consists of physical things equipped with electronic components and ubiquitous intelligence that allow them to connect, interact and exchange data [8], has contributed to the deployment of millions of connected devices in almost any imaginable scenario. Similarly, the Industry 4.0 paradigm has made available multi-sensory data of industrial processes that allow complex algorithms to control and optimize the performance of industrial plants [9].

The current trend is to move data processing from the cloud to the edge. In particular, ML algorithms are being increasingly deployed in embedded devices [10]. There are many reasons why computing at the edge is preferable to computing at the cloud [11]. On the one hand, the amount of data traffic increases together with the number of deployed devices. On the other hand, data transmission and processing in remote systems introduces a delay that in some cases is unacceptable. Additionally, there may be security issues if private or sensitive information needs to be transmitted from local facilities to an external data center [12].

In the literature, edge devices are vaguely defined. Even if the premise is always that the processing is located near the source of the data, this could refer to both a computing network infrastructure located in the same facilities as sensors or an embedded device with a tiny micro-controller. In the present work, edge devices are understood to be embedded devices that usually incorporate sensor data acquisition hardware and are able to autonomously execute data processing algorithms and make some “smart” decisions.

### 2.1. Selection of Embedded AI Hardware Devices

The first challenge to benchmarking the performance of a DL model in an embedded device is to select the appropriate hardware device itself. There are hundreds of hardware devices that claim to have a design oriented to the execution of ML algorithms. In fact, many modern micro-controllers are actually able to run a set of ML algorithms [13,14], but since one of the goals of this work is to deploy machine vision DL algorithms, a powerful enough device should be selected. On average, the number of operations required to compute a complete inference from an input image is around some tens of billions of operations or Giga-Operations (GOPS) [15]. Since a video sequence has around 30 to 60 frames per second, it is estimated that the minimum computational power an embedded device must have is around one Tera-Operations per second (TOPS). This requirement rules out most general-purpose micro-controllers, for example those based on the widely used ARM CortexM architecture, and also many application processors, including those based on the ARM CortexA architecture. Even some processors based on the x86 architecture are not powerful enough. To reach those figures, it is necessary to select a processor with a specific integrated mathematical co-processor. Due to the great success of DL, modern embedded hardware devices have begun to integrate powerful AI co-processors to perform DL computations. There are three main solutions to integrate a DL-oriented co-processor in embedded hardware: (i) use a general-purpose processor that already integrates a co-processor in the same semiconductor die; (ii) include a separate Application Specific Integrated Circuit (ASIC) designed for DL inference together with the general purpose processor in the embedded hardware design; or (iii) use a programmable logic device (CPLD or FPGA) to implement custom co-processor hardware [16]. The design of a math accelerator circuit for DL model inference is outside the scope of this work, and therefore the third solution is rejected in favor of the first two. Based on these criteria, the embedded hardware devices selected for this work are described in the next sub-sections.

### 2.2. NXP i-MX8M-PLUS Application Processor

The first hardware platform selected is the i-MX8M-PLUS processor. It is an NXP heterogeneous multi-core processor for high-performance applications focused on video processing and DL (https://www.nxp.com/products/processors-and-microcontrollers/arm-processors/i-mx-applications-processors/i-mx-8-processors/i-mx-8m-plus-arm-cortex-a53-machine-learning-vision-multimedia-and-industrial-iot:IMX8MPLUS, accessed on 11 July 2021). The embedded System on Chip (SoC) from Variscite shown in Figure 1 and the matching evaluation kit were used in this work.

From a DL application development perspective, the most interesting component of this board is the embedded Neural Processing Unit (NPU) with 2.3 TOPS of computing power. It is also quite remarkable that the NPU is integrated onto the same die as the general-purpose processors and shares the high-speed internal memory bus. This architecture helps speed up the DNN inference as the data interchanged between both computing units are optimized. The NPU is a Vivante VIP8000 specifically designed for being embedded in processors of the i-MX family. It works with 8-bit integer data types (INT8) rather than 32-bit floating-point data (FLOAT32). As will be seen in Section 5, this means that the DNN needs to be transformed (quantized) before being executed in the NPU. NXP provides the entire ecosystem of tools to manage the entire workflow pipeline, including the design, deployment and inference of neural networks. The processor also features a powerful image-processing pipeline, camera interfaces and a comprehensive set of communication peripherals.

### 2.3. Google Coral Dev Board with EdgeTPU Module

The other hardware platform considered in this work is the Coral Dev Board. This is an evaluation kit for the EdgeTPU AI accelerator module (see Figure 2), an ASIC with a PCI or high-speed USB communication interface that performs 4 TOPS while drawing 2 W of power. It also uses INT8 operands, and it is designed to add DNN inference ability to general-purpose processors.

The Coral Dev board integrates an NXP i-MX8-MINI processor from the i-MX8 family designed for industrial applications. It is slightly less powerful than the i-MX8M-PLUS, with fewer image peripherals and interfaces and without the integrated AI co-processor—that role is played by the EdgeTPU. Note that the two devices selected for this work are partially compatible, as both use processors from the i-MX8 family. This was, as a matter of fact, one of the reasons they were chosen. However, Google provides its own tool set for both the EdgeTPU and the i-MX8-MINI SoC, based on a Mendel Linux distribution and TensorFlow Lite framework.

## 3. Deep Learning Frameworks

ML’s success and popularity could not be understood without the existence of powerful and, at the same time, user-friendly application development frameworks. Some technology companies and universities have developed complete ML inference libraries for their own research purposes that they have ended up making public as open source software. Many ML algorithms are based on complex and quite cumbersome mathematical formulations that are not easy to implement. Frameworks simplify the development of such algorithms by exposing a high-level API to deal with complex calculations. In the case of DL networks, frameworks allow the implementation of a complete workflow, including defining the network architecture, training and optimization, model performance testing and model deployment into the final embedded devices.

There are many frameworks to choose from, and in general there are a lot of resources available on the web for almost all of them, but some frameworks have gained popularity among programmers and offer better support for application development. In [17], some of the most popular DL frameworks are classified by user access statistics to GitHub repositories. These frameworks demand considerable computing power, and they run on powerful computers usually complemented with GPUs [18]. Some of the processes involved in DL applications, such as model training and validation, require a large amount of memory and computational power. For that reason, they still run on high-end computing systems, and rarely on embedded devices.

Each framework uses its own model formats and APIs to build and implement DL applications. If the model is going to run in an embedded device, the framework must be supported by the embedded software distribution. This in fact determines the selection of the framework in the host (high-end) computer because the software of the host and the device must be compatible. To deal with this challenge, a standard interoperability library called Open Neural Network Exchange (ONNX) (https://onnx.ai/, accessed on 20 July 2021) was designed. Many embedded software distributions support this standard, allowing the selecting of the host framework without worrying about embedded device compatibility issues, as shown in Figure 3. Furthermore, this means that, at least theoretically, any model developed using any ML framework could be deployed into any embedded device by adequately converting the format of the model. In reality, embedded software distributions present strong restrictions, even more so if the embedded hardware integrates design-specific AI co-processors, so interoperability is far from total. A main issue is that ONNX is not widely supported by all embedded devices, and hardware manufactures provide specific libraries to deploy DNN in their co-processors that support a limited, if not unique, model format. For this reason, in the following sections the frameworks and libraries available in the selected embedded devices are revised.

### 3.1. Yocto Distribution and eIQ Machine Learning Framework for NXP i-MX8M Processors

The Yocto Project (https://www.yoctoproject.org/, accessed on 20 July 2021) is an open-source collaborative project that helps developers create custom Linux-based systems regardless of hardware architecture. NXP (the manufacturer of the i-MX8M-PLUS processor) provides a software release based on the Yocto Project framework. It can be used to build images for any i-MX8M board.

The compilation process downloads and installs many libraries and packages to create the binary image of a functional Linux distribution for the board. This binary image contains all the resources NXP provides to create an embedded ML application. In particular, the eIQ development environment supports these six run-time environments (inference engines): ArmNN, TensorFlow Lite, ONNX Runtime, PyTorch, OpenCV and DeepView${}^{\mathrm{TM}}$RT. To fully exploit the potential of the board, the framework selected must be supported by the internal NPU processor. Figure 4 shows the supported eIQ inference engines across the i-MX computing units.

Pytorch and OpenCV are not supported by the embedded NPU and are directly discarded. A user guide (https://www.nxp.com/design/software/embedded-software/i-mx-software/embedded-linux-for-i-mx-applications-processors:IMXLINUX, accessed on 20 July 2021) explains the capabilities of all inference engines. For reasons that will become apparent in the next subsection, the most suitable runtime environment for this work is TensorFlow Lite (https://www.TensorFlow.org/lite/guide, accessed on 20 July 2021). As the name suggests, this is a lightweight version of the TensorFlow library for mobile, IoT and embedded devices. It is a runtime package that provides a way to run Deep Neural Networks on a specific hardware processor.

### 3.2. Mendel Linux and TensorFlow Lite in Coral Dev Board

The Coral Dev Board uses a Mendel Linux distribution maintained by Google. Unlike NXP Linux distributions, Coral Mendel Linux is specifically designed for this evaluation board kit, so there is no need to configure and compile the kernel or install any software packages or libraries. Everything is already available in a binary image that can be downloaded from https://coral.ai/docs/dev-board/get-started/ (accessed on 20 July 2021). The Coral Dev Board has a complete runtime ready to deploy DL models on its EdgeTPU AI co-processor unit. This co-processor was designed by Google to deploy TensorFlow models in embedded hardware, so the use of TensorFlow and its variant TensorFlow Lite is mandatory. TensorFlow Lite models must be off-line processed with a specific tool named “EdgeTPU Compiler” before being deployed in the EdgeTPU AI co-processor.

### 3.3. Host PC Setup

The host computer is an essential part of the whole development ecosystem. For this work, a host PC running Ubuntu 18.04 64-bit is used. The ML framework installed in the host is TensorFlow 2.5.0. The selection was straightforward, as both embedded devices support the TensorFlow Lite runtime. It comprises many functionalities, but the only one used in this work is the ability to convert object detection models into “lite” formats suitable for embedded systems. The TensorFlow programming interface is mainly written for Python, and it was decided to use this language to write all the model conversion scripts.

TensorFlow (and TensorFlow Lite) can be integrated with Python and C/C++ applications. It was decided to use Python to develop all the necessary scripts for the benchmarks described in this paper.

## 4. Object Detection Models

Object detection models are specialized ANN architectures designed to solve the computer vision task of object identification and localization in a digital image. From the model architecture perspective, object detection models inherit the feature extraction backbone from classification models. It is common to implement an object detection model by reusing a classification model such as VGG16, Mobilenet or Resnet, trained on a very large image dataset. The backbone used in embedded devices must be carefully selected, as the number of layers in the models varies greatly. Integration of the classification and localization heads in the model defines two separate solutions: two-stage models and one-stage models, in reference to the number of functional parts that the model contains. In the case of two-stage models, the first stage generates region proposals for object detection, and the second stage computes each proposed region and extracts both the classification result and the bounding boxes. Compared to one-stage models (which perform all functions together) two-stage models tend to have higher accuracy, although at a higher computational cost [19]. One of the first and most representative two-stage models is R-CNN [20], whose region proposal stage proposes around 2000 regions from the input image.

One-stage models use a feed-forward architecture in which everything is inferred in a single pass by applying a single neural network to the entire image. This approach results in significantly lower accuracy than two-stage detectors, but also higher detection speed. One of the first one-stage detectors was YOLO [21].

The TensorFlow library is accompanied by auxiliary libraries that complement its functionalities. Of particular interest for DL is the TensorFlow models repository (https://github.com/TensorFlow/models, accessed on 30 July 2021), also called the TensorFlow model zoo. This repository contains models for many DL applications, such as natural language processing, speech recognition and object detection. The model git repository version 2.5.0 was cloned (in accordance with the TensorFlow version). Inside the “models” directory, the “official” folder includes the code and models directly maintained by Google. The “research” folder contains some state-of-the-art technologies maintained by the developers themselves. The “object_detection” directory inside the “research” folder contains the libraries, code and models that have been used for hardware benchmarking. A brief explanation and an installation procedure can be found in https://github.com/TensorFlow/models/blob/master/research/object_detection/g3doc/tf2.md (accessed on 30 July 2021). The TensorFlow model zoo contains several types of object detection model architectures, which are described in the following paragraphs.

### 4.1. CenterNet

CenterNet (https://github.com/xingyizhou/CenterNet, accessed on 15 September 2021) is a one-stage object detection network that infers object position by assigning one point to every object rather than a square [22]. The size and even the pose of the object are calculated afterwards using a regression network. This strategy increases the accuracy of the network while maintaining fast inference time.

### 4.2. Single Shot Multibox Detection (SSD)

SSD networks [23] are widely used in embedded devices. They were the first one-stage networks, along with YOLO networks, that achieved accuracy similar to that of two-stage networks. Combined with the “mobilenet” backbone, it is the most supported network in TensorFlow Lite, mainly because it was developed by Google Research (among other researchers from academia) and it is a lightweight network suitable for deployment in embedded devices.

SSD networks usually come with a specialized component named a Feature Pyramid Network (FPN) [24] designed to improve the detection performance with objects at different scales. Usually object detection networks function quite poorly with very small or very big objects (in terms of the number of pixels that an object occupies in the image). FPNs solve this problem, increasing detection accuracy but also increasing processing time.

### 4.3. EfficientDet

The EfficientDet [25] DNN describes an improved one-stage network architecture that can be optimized and scaled to obtain a complete family of neural networks. Depending on the available computing resources and requirements, it is possible to select the most adequate member of the family. EfficientDet-D0 is the least resource demanding network of the family, and it should be adequate for embedded devices. The backbone used as feature extractor is called EfficientNet, hence its name.

### 4.4. Faster R-CNN

Faster R-CNN [26] is a two-stage object detection network. This architecture incorporates a new first-stage region proposal that improves network performance, achieving inference times comparable to those of single-stage networks while maintaining high accuracy. It is the latest of consecutively improved architectures, starting with R-CNN, then Fast-RCNN and finally Faster-RCNN. Some enhancements are also applied to the Faster R-CNN architecture to improve both inference speed and result accuracy [27,28].

### 4.5. Mask R-CNN

Mask R-CNN is an object segmentation model [29]. Object segmentation is a technique that, instead of detecting the object inside the image, categorizes each individual pixel of the image as belonging to a particular class. The goal is to obtain all the pixels belonging to a given class in the image, being able to draw the silhouette and the exact contour of an object, not only the surrounding square. In this sense, object segmentation can be seen as an improvement over object detection. Some architecture enhancements are available in the literature [30].

## 5. Model Conversion for Embedded Hardware Devices

The Design and Training stages of a DL model are almost always accomplished using a powerful host computer. The host computer includes an installation of a full ML framework with a set of packages and libraries to support and facilitate the whole DL application development workflow. The embedded devices, on the other hand, contain a runtime environment designed only and specifically to run a DL model inference.

In the TensorFlow environment, a model is described by a computational graph containing both the node connections and the weights or parameters of each node. The model is usually defined as a code file containing the API function calls necessary to build the model, for example using Keras API (https://keras.io/getting_started/, accessed on 15 September 2021). The model is built sequentially by adding a series of computational layers that fully describe the model architecture. However, at this point, the model is not functional because it does not yet contain the value of the weights, which are computed in the training process. Weights are stored in separated files named checkpoints. A checkpoint can be stored and reloaded at any time. This allows comparing the performance of different training stages, or retraining some of the model layers to accomplish an object detection task different from the one the model was previously trained for. Once the model is created, it is possible to save the computational graph and the weights all together in a single file format named “SavedModel” format using a specific TensorFlow API function call. A brief tutorial on TensorFlow model formats is available in https://www.TensorFlow.org/tutorials/keras/save_and_load (accessed on 11 July 2021).

For the TensorFlow Lite runtime environment, models created in TensorFlow must be converted using a specific library. This process modifies the model format appropriately to adapt it to run efficiently on the specific AI co-processors. Conversions mainly affect model weights, input tensors and output tensors. In general, TensorFlow models by default use floating-point parameters, which are appropriate for high-performance CPUs and GPUs, but embedded AI accelerators normally are restricted to work with integers only. Converting from float to integer types is called quantization.

In this work, five different quantization levels are considered based on the TensorFlow Lite optimization guide (https://www.TensorFlow.org/lite/performance/model_optimization, accessed on 11 July 2021). A brief description of the quantization levels is presented in Table 1, assigning to each level a numerical value. Note that the TensorFlow Lite conversion with no quantization has (properly) a quantization level 0. In the rest of this work, models with quantization levels 0 and 1 will be referred to as CPU models since they will run entirely on the main processor. In contrast, level 2, 3 and 4 models are intended to be executed in the specialized AI co-processor and will be referred to as co-processor models. An important part of this work is to measure the performance advantages of co-processor models over CPU models when an AI accelerator is available.

### 5.1. Model Conversion Issues

The model conversion workflow is depicted as a block diagram in Figure 5. Models downloaded from the TensorFlow model zoo are already trained. The parameters in the trained checkpoint files are exported into a “SavedModel” file, and afterward model conversion is applied. Five conversion Python scripts were implemented to obtain the five corresponding TensorFlow Lite models, one per quantization level. These models are ready to be deployed in the i-MX8M-PLUS processor, but for the EdgeTPU module an extra compilation step must be done using a specific compiler developed by Google named “edgetpu_compiler”. Therefore, after this compilation another five quantized models are obtained.

There are more than 80 models available In the TensorFlow model zoo (https://github.com/TensorFlow/models/blob/master/research/object_detection/g3doc/tf2_detection_zoo.md, accessed on 30 July 2021). Table 2 lists the nine models selected to be used in the present work. The name of each model describes the architecture, the input tensor size and the dataset used for training (all models are trained using COCO 2017 dataset). Some of the models integrate a Feature Pyramid Network (FPN) component, which improves the detection of objects at different scales in the image. Note that all the object detection architectures from the TensorFlow model zoo are represented except for Mask R-CNN. This model is in fact an object segmentation model with very different inference results and computation requirements, not comparable with the others, and for this reason it was not included in the benchmark. The justification of the selection of the rest of the models will become clear in the following subsections. For a given network, a total of ten optimized embedded “.tflite” models are generated (five for i-MX8M-PLUS and another five for EdgeTPU). Considering the nine selected DL models, 90 embedded models were obtained. However, application of the conversion scripts was not always completed successfully. In Figure 6, all the issues found when trying to convert checkpoint files to TensorFlow Lite formats are listed. The next paragraphs explain each of them.

#### 5.1.1. Unable to Export Checkpoint Files

The simplest way to convert a model is to use a “SavedModel” format from TensorFlow. It is possible to download any model from the TensorFlow model zoo in “SavedModel” format and also the training checkpoint files. Once uncompressed, it contains the saved_model directory with the “.pb” file, together with the checkpoint directory with checkpoint files. There is also a configuration file “.config” describing the model architecture. Unfortunately, this default “SavedModel” format is not suitable for conversion to TensorFlow Lite format because all object detection models have internal operations not supported by TensorFlow Lite. Instead, the object detection model code library provides a specific Python script to create a valid “SavedModel” from checkpoint files called “export_tflite_graph_tf2.py”. However, this script supports neither EfficientDet nor Faster R-CNN network architectures. This means that approximately one half of the networks in the model zoo are in fact not suitable for use in embedded devices.

Some alternative model repositories were reviewed to try to overcome this problem, but the models must indeed fulfill so many constrains to be used with TensorFlow Lite that in the end only TensorFlow models were valid. A Lite version of “EfficientDet” model already converted to “.tflite” format was found at https://tfhub.dev/TensorFlow/efficientdet/lite0/detection/1 (accessed on 15 September 2021). It was also compiled for EdgeTPU and was included in the benchmark with the name “Efficientdet_lite0_320”.

#### 5.1.2. Experimental CenterNet Model Export

CenterNet models checkpoint export fails when using TensorFlow library versions older than 2.4.0. Starting with this version, support for these networks was added. However, the export command requires small modifications compared with the command provided in the TensorFlow model optimization guide. The specific conversion command should be consulted in an example Jupyter Notebook at https://github.com/TensorFlow/models/blob/master/research/object_detection/colab_tutorials/centernet_on_device.ipynb (accessed on 15 September 2021). In the export command, the model size is also modified from its original value to 320 × 320. The model name is modified to reflect this change in the figures herein.

#### 5.1.3. EdgeTPU Compiler Fails

Co-processor models of “ssd_resnet101_v1_640_fpn” could not be compiled to EdgeTPU format. The compiler does not provide any information about the reasons for this failure. The network is by far the largest in the benchmark (more than 200 MB), so it is assumed that it in some way exceeds the capacity of the EdgeTPU module (or of the compiler itself).

### 5.2. Converted Model Size Analysis

Three files describe each model before the conversion: checkpoint file, original saved model file and exported saved model file, obtained from the checkpoint file after the execution of “export_tflite_graph_tf2.py”, as explained above. Figure 7 displays the size of such files. The size range is from approximately 20 MB to more than 220 MB. The exported saved model file and the original saved model are similar, with the former slightly bigger than the last, and the checkpoint file is some MB smaller than the other two, except for CenterNet network. All other networks have a single-shot detection “SSD” architecture, and this could explain the difference.

The models are sorted by ascending size of the exported file (gray column in the figure). There are no TensorFlow files for “EfficientDet” network, so it was positioned in its corresponding position, attending to quantization level 3. This model order will be maintained in the rest of the document.

The converted TensorFlow Lite model file sizes are shown in Figure 8 for i-MX8M-PLUS and in Figure 9 for EdgeTPU. The names of the quantized model files start with a number indicating the quantization level. The converted model without quantization (level 0) is smaller than the original model when the model itself is small, but exceeds the original model size considerably for the largest models. The other converted files present some kind of optimization. Starting from quantization level 1, model files present a type conversion of the network weights. Its size is, as expected, four times smaller than the model without quantization. Level 2, 3 and 4 models are slightly larger than those of level 1 (some hundreds of kilobytes) to include quantization of the inner intermediate layers and activation functions. There is no significant difference between the converted models for i-MX8M-PLUS and EdgeTPU devices.

## 6. Embedded Hardware Benchmarks

Once the embedded models are created, the next step is to execute them on the embedded hardware devices. To run the inference, a Python script is implemented using the TensorFlow Lite runtime environment API. Each of the quantized models is slightly different from the rest, so it is mandatory to write ten different Python scripts to execute all of them. The benchmark has two parts: (1) verification of the correctness of the model inference by examination of the obtained results, and (2) measurement of the models computation times. Three computation times are of interest when measuring the performance of the selected hardware devices:**Warm up time**. This is the time the devices use to initialize their specific AI co-processor. Usually, the first inference is used for this initialization in addition to the inference itself. The device is not functional until the warm-up finishes.**Auxiliary (image) processing time**. This is the time the CPU needs to access the image, resize and maybe re-scale it, load it into the input tensor and, after inference, get results and store a new image with bounding boxes around the detected objects.**Model inference time**. This accounts for the time used to execute the mathematical model’s operations, from the input tensor initialization to the access of the output results. Ideally, all the model operations should belong to the AI co-processor, but actually, due to limitations in model conversion and model deployment, some operations are delegated to the general-purpose CPU.

### 6.1. Model Inference Issues

Many issues were identified during the benchmark test. The following sections explain each of the errors or malfunctions detected, pointing out which models fall into each category. Figure 10 collects all of them.

#### 6.1.1. Unable to Execute the Model

In this category two types of issues arise. In the first type, there is no model to be tested because it was not possible to create it. This is the case for EfficientDet and Faster-RCNN models, which are not supported by the export script “export_tflite_graph_tf2.py”; the same applies to the co-processor models for the “SSD_Resnet101” model for EdgeTPU. The second type of errors affects i-MX8M-PLUS with SSDResnet models that are not quantized (level 0). They have the biggest size of all models, and in addition, since they are not deeply optimized, they are executed almost completely in the CPU. This exceeds the memory or hardware capacity of the i-MX8M-PLUS processor and results in a fatal execution error.

#### 6.1.2. Bad Inference Results

In the case of “SSD_Mobilenet” models, the objects detected by level 4 models have wrong class and position values. In this optimization level, the output tensor is converted to INT8 type. The level 3 model with FLOAT32 output tensors behaves correctly, so the error may be due to the quantization process or even to an internal quantization factor that is not being taken into account.

Similar behavior is observed in the case of “SSDResnet” co-processor models for i-MX8M-PLUS. However, in this case, the detection scores are also very low (<10%), indicating that the problem is even worse.

Finally, some models do not detect any objects. This is the case for all co-processor models for CenterNet in both devices and for “SSDResnet50” models only in the EdgeTPU module. Figure 11 shows a bad inference output results image.

#### 6.1.3. No Inference Time Improvement

The level 2 and 3 models for “SSDMobilenet_V1” in EdgeTPU detect correct object classes and position, but they present almost the same inference time as the CPU models. Currently, the EdgeTPU compiler cannot partition the model more than once, so as soon as an unsupported operation occurs, that operation and everything after it executes on the CPU, even if supported operations occur later. See https://coral.ai/docs/edgetpu/models-intro/ (accessed on 15 September 2021) for a more detailed explanation. This could explain this anomalous behavior.

#### 6.1.4. Input Tensor Value Range

The input tensor value range is not the same for all network models. The “SSD_Mobilenet” networks present a FLOAT32 range of [−1, 1] and a quantized UINT8 input of [0, 255]. All the other networks have the same [0, 255] input range for both float and integer models. This supposes a small modification in the inference script for quantization levels 0, 1 and 2 for models with float input tensors.

#### 6.1.5. Good Inference Results

It is not difficult to identify an incorrect behavior described in previous paragraphs because the errors are very evident. However, in general, inference results vary slightly among models and even among quantization levels. Usually, some models detect some object in an image that other models do not detect but fail to detect an object in another image. The detection scores vary from model to model and, because a limit score of 50% was imposed in the test, the objects near the limit may or may not be detected. Inference results are measured by visual inspection rather than by using a function that calculates the possible error. If these results are satisfactory, it is understood that the model is globally correct. Figure 12 shows good inference results for some test images.

### 6.2. Analysis of the Computation Times

All the benchmark tests were conducted using the same image dataset of twenty images taken from the COCO set. The inference script loops for each image in the dataset and stores the computation times. The average values of all computation times are analyzed in the next sections.

#### 6.2.1. Warm Up Time Analysis

Warm up times for the i-MX8M-PLUS are displayed in Figure 13. The figure shows clearly how the warm-up time increases with model size. It is also evident that the co-processor models present much larger times than the other CPU models. This could be easily explained by taking into account that the latter are executed completely in the CPU, so AI co-processor initialization is not necessary, while the former are deployed in the AI co-processor.

The warm-up times vary for co-processor models from approximately 10 s to about 150 s. For small, non-quantized models it is smaller than 10 s, but when model size increases, the warm-up time is extremely long. In fact, the largest model raises an execution error. Quantization level 1 presents warm-up times from some seconds to around 25 s. All these figures represent a considerable amount of time, which must be considered in application design and development.

In the EdgeTPU module, the warm-up times behave differently than in the i-MX8M-PLUS (see Figure 14). The warm-up time for co-processor models is nearly the same as that of any other inference time, showing no significant overhead in EdgeTPU module initialization. For small models, the warm-up time is in the order of hundreds of milliseconds, making a specific initialization stage unnecessary. However, the EdgeTPU did not behave well when the model size increased, showing warm-up times of more than 10 s. Indeed, the largest co-processor models do not run in the EdgeTPU module.

#### 6.2.2. Auxiliary Processing Time Analysis

Auxiliary processing times are fairly homogeneous in all network architectures. For i-MX8M-PLUS (Figure 15), the values vary between 20 and 40 ms with no correlation with model size. However, correlation with model quantization level is observed. The models with float input tensors (levels 0, 1 and 2) present notably larger times than those with quantized INT8 input tensors. This is more evident in “SSD_Mobilenet” networks. It is also observed that in the models with a large input size of 640 × 640, the difference is even bigger. The explanation is straightforward. The “SSD_Mobilenet” models need a preparatory scale operation (those models have a float [−1, 1] input range) that involves floating-point operations in the input image. The cost of these operations increases with the size of the input tensor. The difference ranges form 4–5 ms for 320 × 320 input tensors up to 15 ms for sizes of 640 × 640. This time difference is not very high, but, especially in real time applications, should not be neglected.

Auxiliary processing times in the EdgeTPU are slightly larger (around 5 ms) than those in the i-MX8M-PLUS due to the slightly smaller computing power of the Coral Dev general purpose processor. However, the times behave exactly in the same way as explained above.

#### 6.2.3. i-MX8M-PLUS Inference Time Analysis

The DL model inference time is the most relevant parameter to be analyzed in order to measure the performance of the embedded hardware and the feasibility of the deployment of DL object detection applications. Both devices’ inference times are analyzed independently, starting here with the i-MX8M-PLUS processor, and the results are compared afterwards.

The inference times for the i-MX8M-PLUS strongly depend on quantization level. As expected, CPU models have considerably longer inference times than co-processor models. CPU models’ inference times in Figure 16 range from 500 ms to around 25 s. The quantization level 0 inference time for “SSD_Mobilent_V1” presents an outlier value exceeding one minute. This points to even longer inference times for “SSD_Resnet” networks, but those models do not work on the i-MX8M-PLUS. The co-processor models’ inference times in Figure 17 range from 20 ms to near 800 ms. Note that the timescale in the figure is 100 times lower than in the previous figure above. The yellow line in the figure represents the quantization level 3 models’ inference time and is used later to compare results between hardware devices.

Attending to the inference times, it is clear that “ssd_mobilenet_v2_320” should be moved to first place, and “ssd_mobilenet_v2_640 × 640” should be move back one position behind “efficientdet_lite0_320”. This means that the inference times cannot be directly inferred from model size; rather, network complexity should be taken into account. Sorted by ascending inference time, “SSD_Mobilenet_V2” is followed by networks with Feature Pyramid Network (FPN), which introduces computation complexity, and afterward the models with size 640 × 640 are positioned as expected at the end. It is important to note that there is no significant difference in the inference times between co-processor models with different quantization levels.

Note also that even if they appear in the figure above, CenterNet and “SSD_Resnet” Network do not obtain good inference results. The inference time figures were included in the benchmark because the CPU models worked properly, and the obtained inference times are also coherent with model size and complexity.

#### 6.2.4. EdgeTPU Inference Time Analysis

Inference times for the EdgeTPU module behave nearly in the same way as those of the i-MX8M-PLUS. The times for CPU models (Figure 18) are considerably longer than those for co-processor models (Figure 19). However, the CPU models did not present the anomalous behavior for large models, and all of them were correctly executed on the Coral Dev Board.

In the case of co-processor models, for large models, there is no time reduction compared with CPU models, and those models are omitted in the inference time analysis. The yellow line in the Figure 19 belongs to the quantization level 3 models, as was the case for the i-MX8M-PLUS. The fastest model is, as in the case for the i-MX8M-PLUS processor, the “ssd_mobilenet_v2_320” model, with inference time below 20 ms. The “eficiendet_lite0_320” model, with 145 ms inference time, overtakes the “centernet_Mobilenet_320”, with more than 500 ms, and “ssd_mobilenet_V2_640”, with 650 ms inference time.

#### 6.2.5. i-MX8M-PLUS vs. EdgeTPU Inference Time Comparison

A performance improvement factor is calculated by dividing the inference times of the quantization level 1 model by the inference time of the corresponding model with quantization level 3. The improvement factor for the i-MX8M-PLUS processor increases monotonically with model size, as can be observed in Figure 20. Its value varies from 5 for smaller models up to more than 30 for the largest model, “ssd_resnet_101_V1”.

For the EdgeTPU module, the performance improvement factor presents a value of around 4, except for the network “ssd_mobilenet_v2_320”, which obtains a value of 23. The values are below those of the i-MX8M-PLUS processor, and these results are even worse taking into account that the inference times for quantized level 1 models in the Coral Dev board are longer (around 10%) than the corresponding values in the i-MX8M-PLUS processor due to the computing power differences in the general purpose ARM CPUs of both devices.

In Figure 21, the inference times for quantization level 3 models for both devices are displayed. In the case of the EdgeTPU, only the first, small models are depicted because the last three models do not have valid inference times. The i-MX8M-PLUS processor shows better performance than the EdgeTPU Coral Dev board for the first three models and almost the same performance for the next two. Taking into account that the EdgeTPU has 4 TOPS computing power and the i-MX8M-PLUS has 2.3 TOPS, these results suggest that the i-MX8M-PLUS processor is more efficient than the EdgeTPU module when deploying and running DL models.

This better performance is confirmed by looking at the behavior of the largest models. In the i-MX8M-PLUS processor, the inference time is kept under one second, with a improvement factor of up to 30, while the EdgeTPU module presents times over 10 s and improvement factors below 2.

## 7. Conclusions

The first effect related to AI at the edge paradigm is the emergence of many embedded devices with specialized AI co-processors to execute deep neural network inferences. In this work, after a detailed review of the available embedded hardware devices, two of them were selected to demonstrate and evaluate the feasibility of the deployment of DL object detection models in resource constrained devices: Variscite i-MX8M-PLUS Board and EdgeTPU Coral Dev Board. Requirements to select a device for this analysis included: (1) it must belong to an important and reliable manufacturer, and (2) it must offer a strong development community supporting the tools and applications. The devices selected were designed by NXP and Google. NXP is one of the most successful industrial processor manufacturers, and Google could be the most important player in the AI arena. A large portion of this work was devoted to setting up the hardware devices—understanding what libraries and packages needed to be installed and the appropriate tools to use. One of the main goals of the work was to learn and understand the workflow of AI application development, and it can be concluded that the success of this task depends considerably on the selection of the development framework.

The AI framework used to develop and deploy DL networks in embedded devices was TensorFlow, together with TensorFlow Lite. As a first workflow stage, TensorFlow models need to be converted into TensorFlow Lite format. Even if an easy-to-use tool is provided by TensorFlow Lite to convert the models, the conversion is not trivial because of a number of incompatibilities between both frameworks. Many mathematical operations deeply hidden in the layers of the neural networks are not supported by the Lite version runtime, and the conversion of many model architectures remains still unsolved.

All four main model architectures for object detection in the TensorFlow model repository were considered: “CenterNet", “SSD", “EfficientDet” and “Faster R-CNN”. However, in the early stages we realized that TensorFlow Lite conversion of some of the models was impossible. As a matter of fact, only “SSD” and “CenterNet" architectures are compatible with the current TensorFlow Lite converter; thus, a set of seven models were finally selected: six “SSD” with different feature extractor backbones and one “CenterNet”. Further, an “EfficientDet” model already converted to TensorFlow Lite format was added to test as many architectures as possible.

AI co-processors are very specialized hardware units that only accept eight-bit integers as operands, so the models must also be quantized. Five quantization levels were defined in accordance with the capabilities of the TensorFlow Lite library API. After executing model quantization scripts, 35 models for each device were compiled, plus the 2 already converted, giving a total of 72 models.

It is not easy to understand the quality of the converted model to guess how the model should be deployed in the AI co-processor. As a guideline, in the case of the i-MX8M-PLUS, the inference script returns a list of unsupported operations in the initial execution stage, while in the case of the EdgeTPU, a log file is created when the TensorFlow Lite model is compiling, with the number of operations mapped to both the EdgeTPU and the CPU.

The benchmark consisted of executing all the converted models, verifying correct behavior and measuring the model inference time. Many issues were detected during this process. Some converted models did not detect the validation image objects the same was as the original model; others simply did not run in the embedded devices. The number of models with correct behavior was considerably shortened. Only forty of the initial seventy-two models provided acceptable results. If only quantized models with representative datasets are considered, the number decreases to only 16 models, 2 of them belonging to an “EfficientDet_lite0” network not created by the “standard” workflow. Finally, only the four “SSD_Mobilenet” frameworks were proven to be valid for embedded devices. Again, the problems rely on the efficiency and quality of the converted models and the ability of the embedded runtime to fit the models into specialized hardware.

Both hardware devices, the i-MX8M-PLUS and the EdgeTPU, were able to execute the quickest object detection models in approximately 20 ms. The auxiliary CPU processing time spent another 25 ms. The whole inference time supposes nearly 50 ms, or 20 frames per second. The inference times increased up to 100 ms for more complex network models and even more to 500–800 ms when input image size increased. Even if the EdgeTPU claims to have almost double computing power, this benchmark demonstrates that the i-MX8M-PLUS device performed slightly better in general. The performance improvement of co-processor models compared with CPU models is about 10 times in the i-MX8M-PLUS and 5 or even worse in the EdgeTPU.

A few quick calculations were carried out to determine the quality of the AI co-processor inference time results. The i-MX8M-PLUS processor integrates four ARM Cortex-A53 cores at 1.8 GHz. Assuming (to obtain a very raw estimate of computing power) that the cores are able to execute one operation per clock, the maximum theoretical computing processing power should be around 10 Giga-operations per second (GOPS) for floating-point operations. Compared to the AI co-processor’s 2.3 TOPS, the theoretical optimal improvement factor should be in the order of 100. The calculation is based on very imprecise and simplified assumptions, and the actual number should be lower than the theoretical number. Even though, the improvement factor of 5 to 15 obtained for most of the small “SSD_mobilent” networks is quite far from that figures. Once again, the converted model is not competent to be efficiently executed in the AI co-processor. The models are partitioned when unsupported operations are found, and many operations are delegated back to the general purpose CPU, slowing down the total inference performance.

In general, the feeling about the current state of object detection for embedded devices is that many aspects of performance depend on the efficiency of the software frameworks on both the host computer and the embedded device, and on their ability to extract maximum performance from the embedded hardware co-processors. Those libraries are now under construction and continuous modifications. Nearly every month, NXP releases a new version of the Yocto framework for the i-MX processor family (at least two new versions were released since the first benchmark test was accomplished). Coral also releases new compiler tools, API libraries and trained models periodically. In the case of TensorFlow and TensorFlow Lite, even if the libraries were updated many times along the development of the benchmark, new releases are now available to be downloaded. The repository of models is updated every day (there are continuous commits to the research repository), and an official version is released synchronized with every TensorFlow release.

## 8. Future Work

It should be clear after reading the previous sections that many issues remain open and unsolved. The present work does not make a quantitative assessment of the (numerical) performance of the converted models. Performance correctness is decided by visual inspection of the detected objects and correct object classification. Even if this approach easily detects catastrophic failures (such as those shown in Figure 11), subtle performance variations are undetected. A means to measure the error should be included as part of the inference script. There is a straightforward error computation standard defined by the COCO dataset, called mean average precision (mAP), specifically defined for object detection. This error metric is in fact available in TensorFlow, but needs to be implemented from scratch in embedded devices. It would be interesting to investigate whether different levels of quantization introduce noticeable errors, or whether certain network architectures are more sensitive to quantization processes. We plan to carry out a quantitative evaluation of these aspects in a future paper.

One of the main constraints imposed on the work was the requirement of using pre-built models from the TensorFlow model zoo. TensorFlow provides the possibility to implement the model using a flexible API at different levels of abstraction. It would be illustrative to build the standard object detection models used in this work, or even other similar ones, and to investigate how those models behave after quantization in the embedded devices considered here. The final objective should be to learn if there is a way to optimize model deployment by defining model internal operations and layer connections using supported operations of the TensorFlow Lite embedded runtime. Furthermore, additional model sources besides TensorFlow should be investigated. The ONNX model exchange should allow the import of models from other AI frameworks. The EdgeTPU is only supported by TensorFlow Lite runtime libraries, but the i-MX8M-PLUS has some other supported frameworks, such as DeepViewRT, armNN or the previously mentioned ONNX.

Finally, more hardware devices should be considered. The two embedded boards considered in this work shared many hardware specifications. Both have an NXP i-MX family processor, integrate an integer tensor processor and rely on TensorFlow Lite libraries as a runtime. In order to have a more global view of the hardware performance, different types of embedded devices should be tested. At the beginning of the present work, a third hardware platform called Jetson Nano was pre-selected to be included in the benchmark. The Jetson Nano Nvidia AI platform integrates a floating point arithmetic AI co-processor and uses other specialized libraries called TensorRT. The board was successfully launched, and some preliminary tests have been performed, but the software framework is quite different from the one used with the other two boards, and significant work is needed to implement the inference processes.

## Figures and Tables

**Figure 1 sensors-22-04205-f001:**
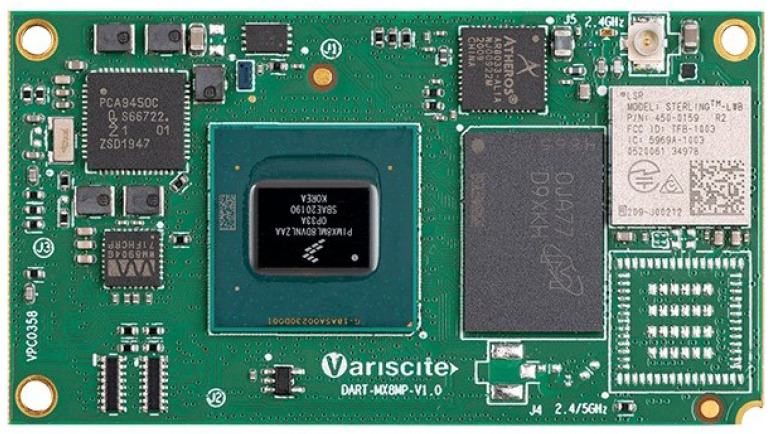
iMX 8M Plus System on Module. Image from https://www.variscite.com/ (accessed on 2 September 2021).

**Figure 2 sensors-22-04205-f002:**
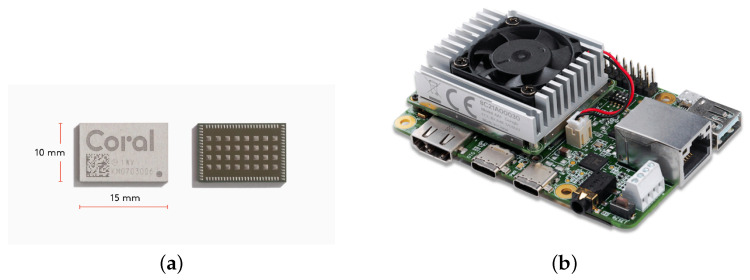
(**a**) EdgeTPU AI accelerator module; (**b**) Coral Deep Learning embedded hardware with EdgeTPU AI accelerator module. Images from https://coral.ai/products/dev-board/ (accessed on 2 September 2021).

**Figure 3 sensors-22-04205-f003:**
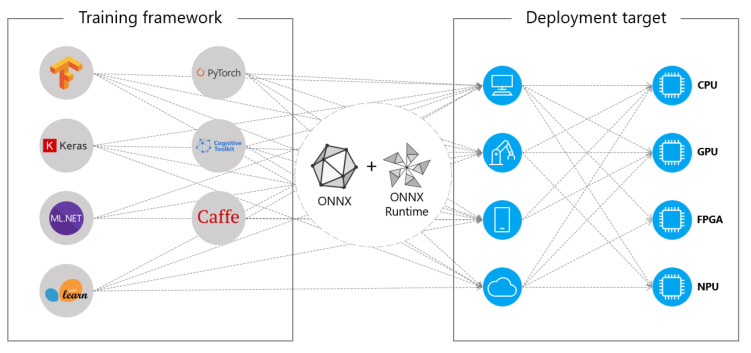
Interoperability of different frameworks by using ONNX.

**Figure 4 sensors-22-04205-f004:**
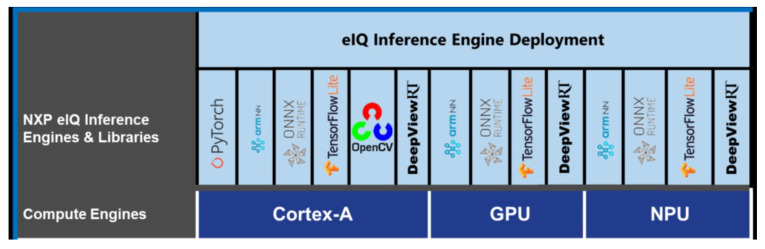
i-MX8 Deep Learning runtime environments supported by embedded computing units.

**Figure 5 sensors-22-04205-f005:**
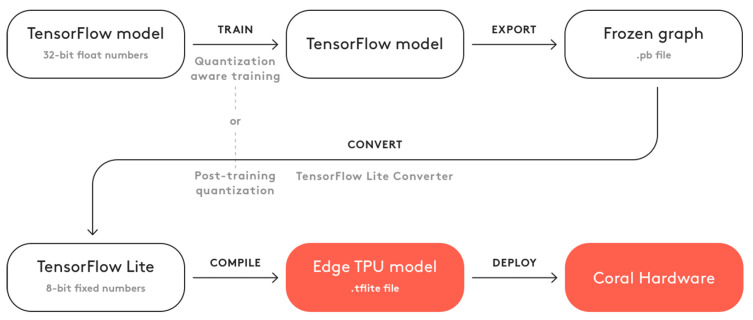
DL model conversion workflow using TensorFlow and TensorFlow Lite.

**Figure 6 sensors-22-04205-f006:**
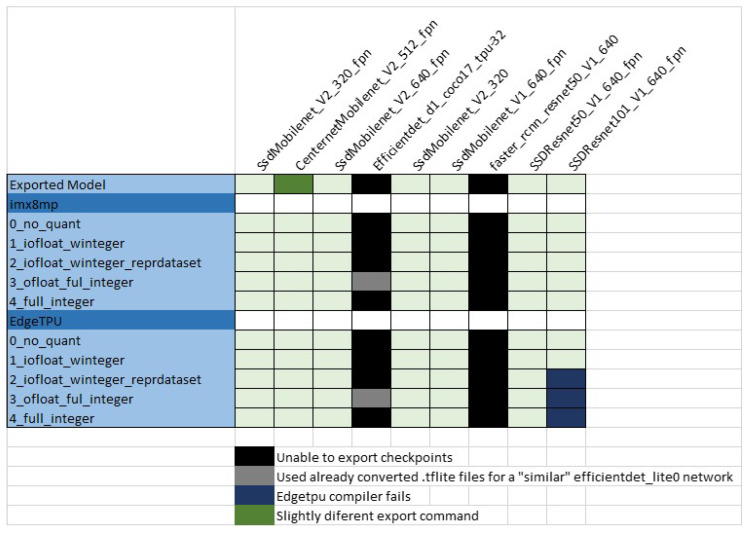
TensorFlow Lite model conversion issues.

**Figure 7 sensors-22-04205-f007:**
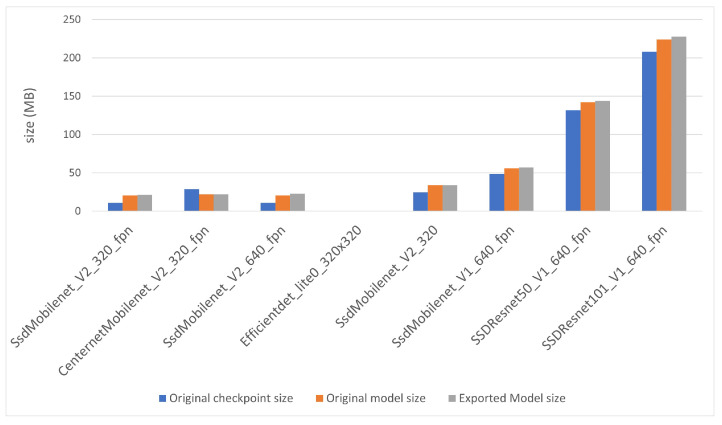
TensorFlow file sizes for object detection models.

**Figure 8 sensors-22-04205-f008:**
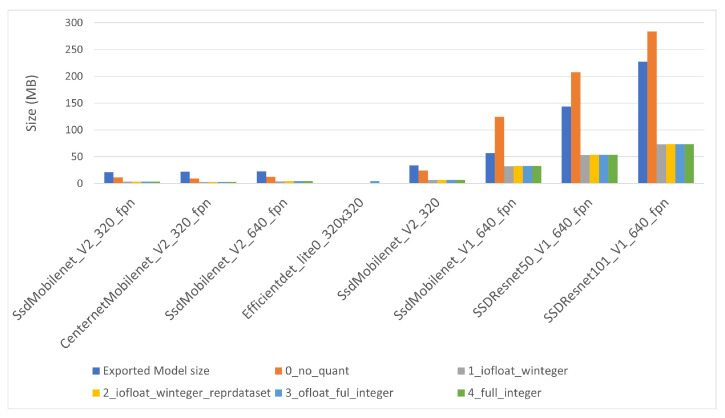
TensorFlow Lite converted file sizes for i-MX8M-PLUS.

**Figure 9 sensors-22-04205-f009:**
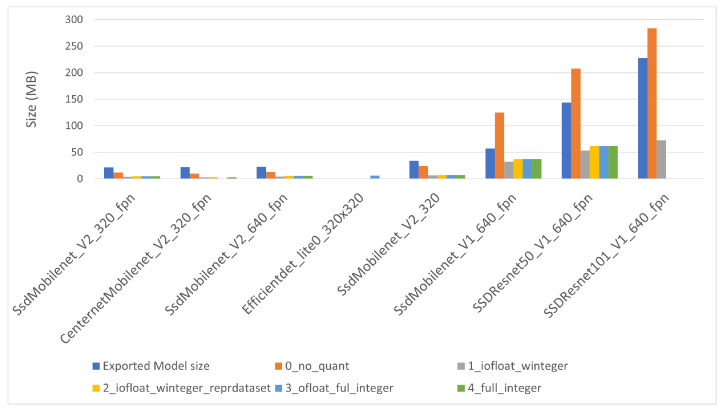
TensorFlow Lite converted file sizes for EdgeTPU.

**Figure 10 sensors-22-04205-f010:**
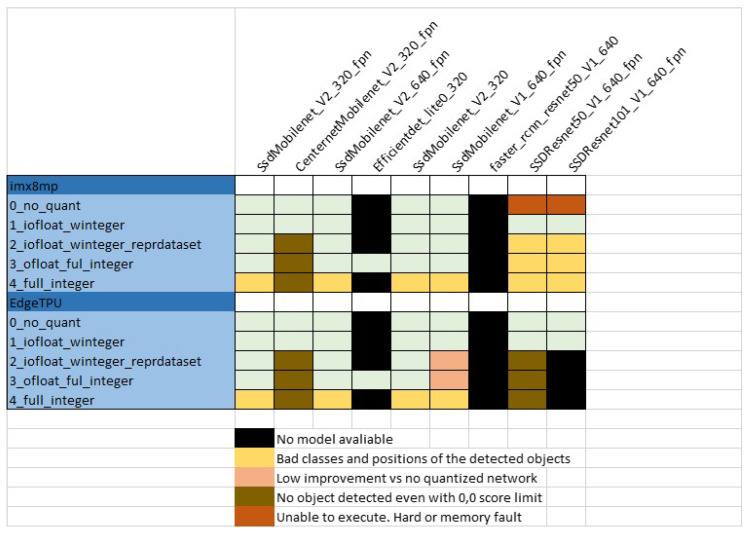
Inference issues.

**Figure 11 sensors-22-04205-f011:**
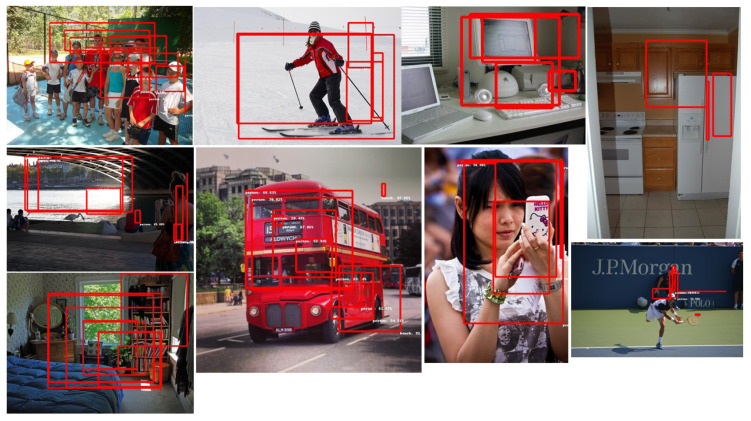
Bad inference results. Neither object location squares nor object class labels are correct.

**Figure 12 sensors-22-04205-f012:**
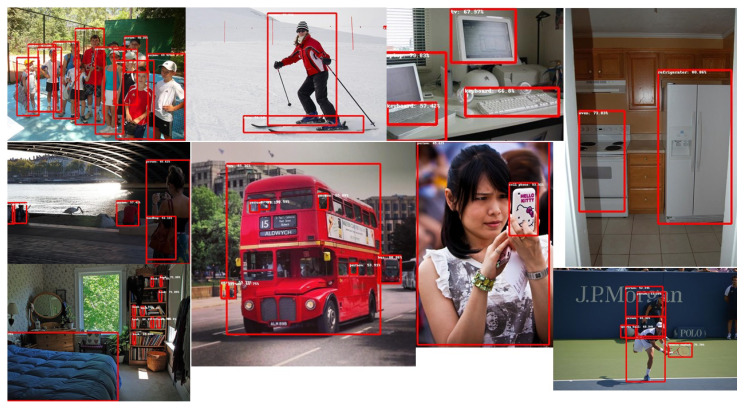
Good inference results.The squares correctly locate object positions and object labels correctly identify object classes.

**Figure 13 sensors-22-04205-f013:**
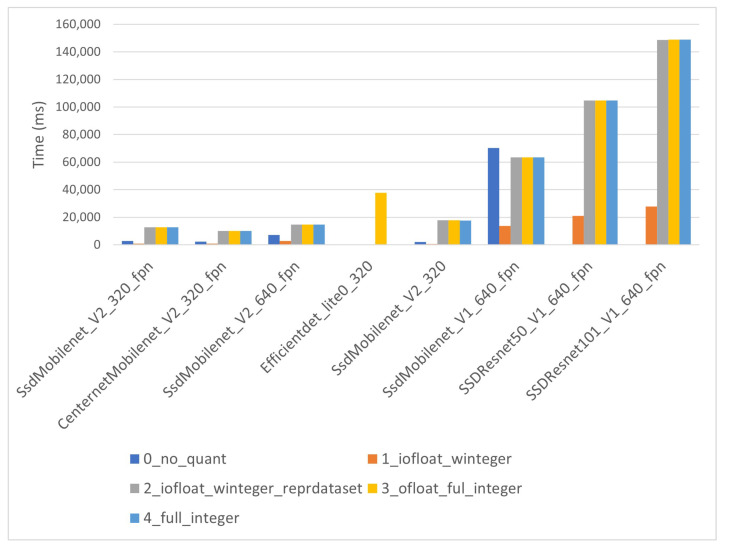
i-MX8M-PLUS warm up times.

**Figure 14 sensors-22-04205-f014:**
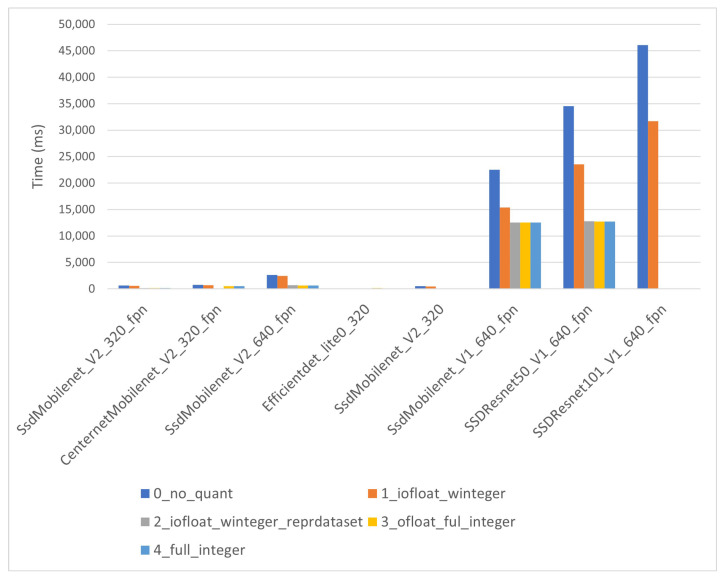
EdgeTPU warm up times for large models.

**Figure 15 sensors-22-04205-f015:**
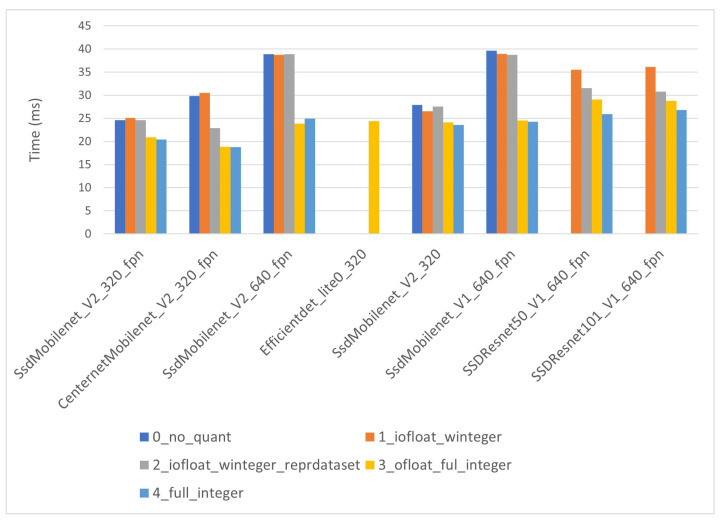
i-MX8M-PLUS auxiliary processing times.

**Figure 16 sensors-22-04205-f016:**
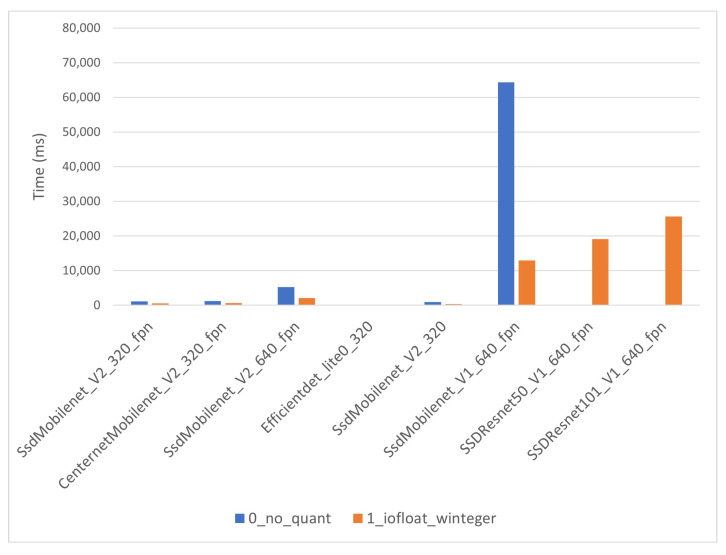
i-MX8M-PLUS inference time for CPU models.

**Figure 17 sensors-22-04205-f017:**
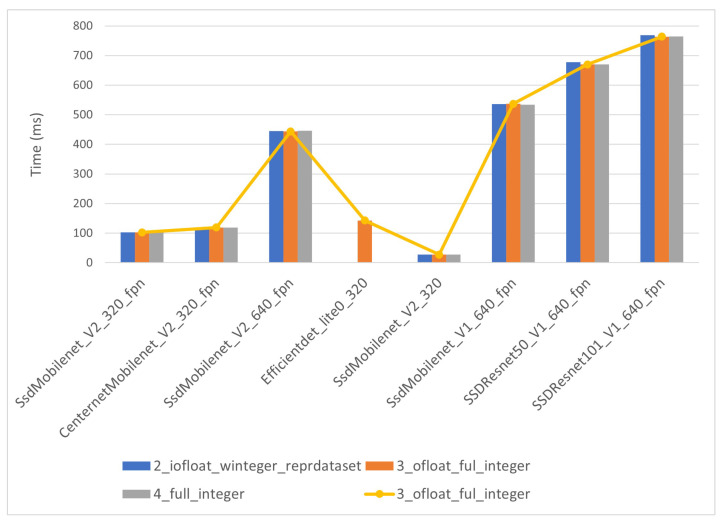
i-MX8M-PLUS inference for co-processor models.

**Figure 18 sensors-22-04205-f018:**
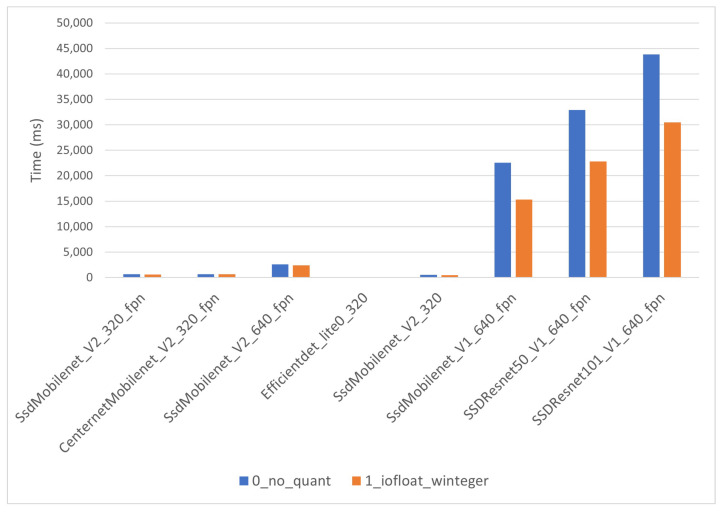
EdgeTPU inference time for CPU models.

**Figure 19 sensors-22-04205-f019:**
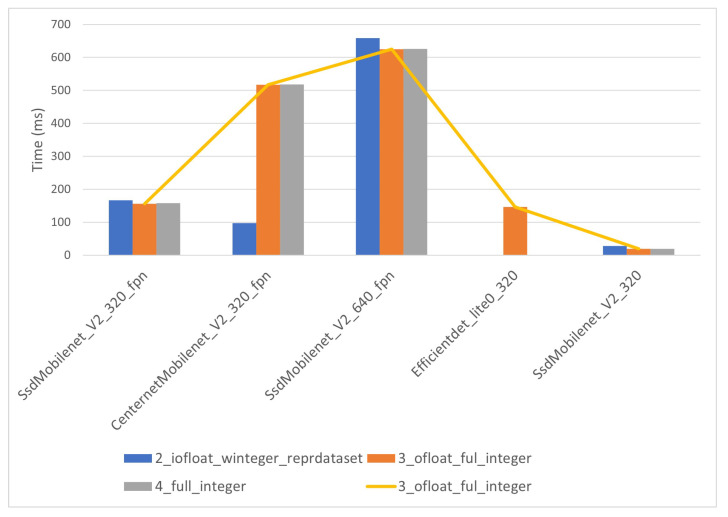
EdgeTPU inference time for co-processor models.

**Figure 20 sensors-22-04205-f020:**
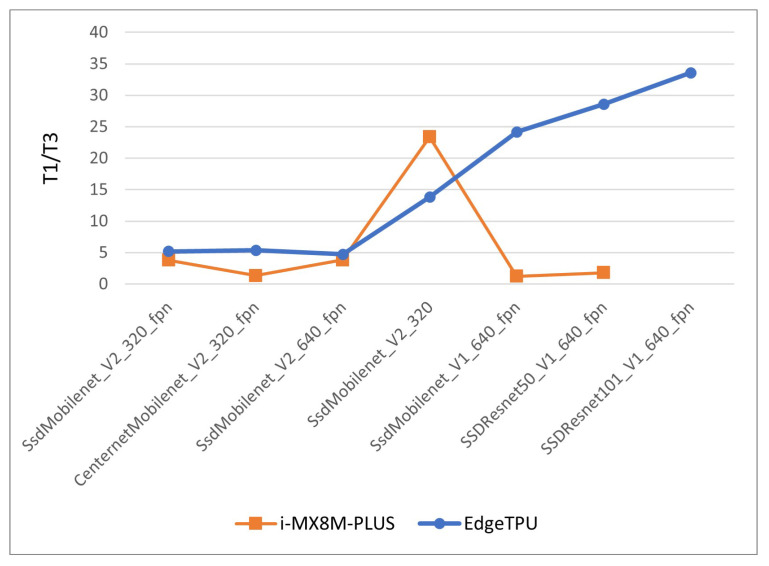
Inference time improvement factor calculated using quantization levels 1 and 3.

**Figure 21 sensors-22-04205-f021:**
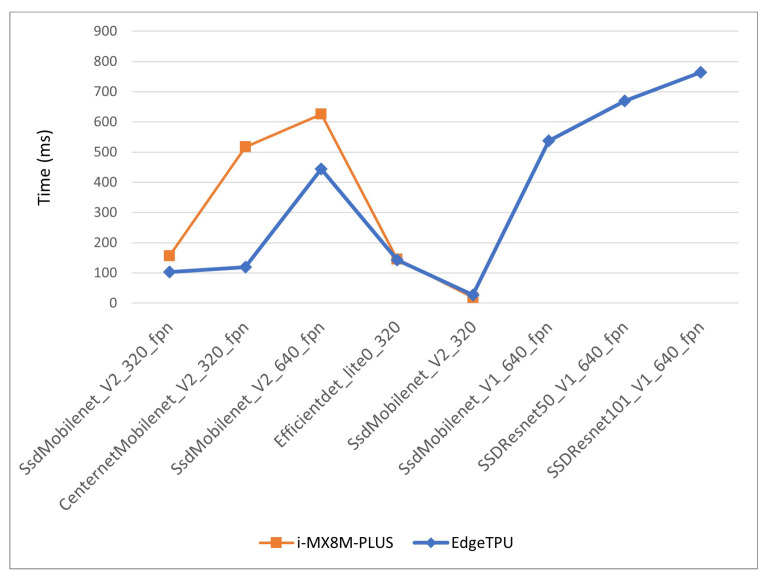
i-MX8M-PLUS vs. EdgeTPU inference times for quantized level 3 models.

**Table 1 sensors-22-04205-t001:** Model quantization (optimization) levels used in this work.

Level	Input	Weights	Output	Description
0	float	float	float	No quantization (all data is FLOAT32)
1	float	int8	float	Quantization of model weights
3	float	int8	float	Quantization of weights and internal variables using a representative dataset. Input and output layers remain in FLOAT32
3	int8	int8	float	Quantization of input tensor uses the representative dataset
4	int8	int8	int8	Full integer conversion. All computation is intended to be done in embedded AI co-processor

**Table 2 sensors-22-04205-t002:** Models used in the hardware benchmark.

No.	Model Name
1	ssd_mobilenet_v2_320x320_coco17_tpu-8
2	centernet_mobilenet_v2_fpn_512x512_coco17_od
3	ssd_mobilenet_v2_fpnlite_640x640_coco17_tpu-8
4	efficientdet_d1_coco17_tpu-32
5	ssd_mobilenet_v2_fpnlite_320x320_coco17_tpu-8
6	ssd_mobilenet_v1_fpn_640x640_coco17_tpu-8
7	ssd_resnet50_v1_fpn_640x640_coco17_tpu-8
8	ssd_resnet101_v1_fpn_640x640_coco17_tpu-8
9	faster_rcnn_resnet50_v1_640x640_coco17_tpu-8

## Data Availability

Not applicable.

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
