# Peer review of "Benchmarking Object Detection Deep Learning Models in Embedded Devices"

_sensors, 2022, doi:10.3390/s22114205_

Round 1

Reviewer 1 Report

Benchmarking Object Detection Deep Learning Models in Embedded Devices

Sensors 

sensors-1724058

The proposed work focuses on object detection aqs an essential capability for robotic applications. It presents a benchmarking study of deep-learning approaches for embedded systems, and the suitability criteria for each.

The paper provides significant technical overview and insight on the tools that are required to achieve a specific goal (object detection) and evaluates the capacity of application from the perspectives of (a) developers and (b) available hardware resources.

The paper provides significant analysis of the field and, in my opinion, merits publications.

Comments:

What comprises an “object” is not clearly defined. My understanding is that humans are also treated as “objects” in this paper. But in this case, there is no distinction between rigid objects and deformable “objects”, where the former task is usually much simpler. This aspect might have an impact on the selection of platform or hardware, but it is not discussed.

Though it is clear from the standpoint of an expert, that deep learning is essential particularly for embedded solutions to object detection, this may not be clear for readers of less expertise. My recommendation, for making this work more “stand-alone” would be to provide a small explanation of why (a) deep-learning approaches are preferred over conventional and (b) that the training of these approaches takes place at different hardware than their execution. Though this is a common background for experts in the field, this information might not be clear to any reader. It should be simple to add a small paragraph in the introduction to accommodate readers with less expertise on the topic.

Reviewer 2 Report

(1)The authors should highlight their works and contributions in Abstract. Why is this work important? Is this work difficult?
(2)The characters in some figures, such as Fig. 5, are too small.
(3)More SOTA methods can be cited and tested, such as:
-Mask Refined R-CNN: A network for refining object details in instance segmentation
-Object detection based on multi-layer convolution feature fusion and online hard example mining
(4)The length of this paper is a little bit long. Some well-known knowledge can be removed or shortened, since they can be easily found in textbooks, and are not firstly proposed in this paper.
(5)In some figures, such as Figs. 7~9, it is difficult to observe the sizes of Efficientdet_lite.
(6)EdgeTPU is supported by TensorFlow Lite, but i-MX8M-PLUS can supported by other framework. How about other frameworks?

Round 2

Reviewer 2 Report

The authors' contribution is mainly to realize the model conversion and the transplantation to the embedded device, but the indicators obtained by the current tests are unclear. For example, the inference threshold is only 0.5, and the confidence level of the detected target in the figure, which is used to show the good inference results, is generally around 0.6. This score is not compared with other transplantation methods, and the inference results of the untransplanted (original) model are absent. Thus it is difficult to intuitively judge whether the inference results are excellent.
